# Global Hemostasis Potential in COVID-19 Positive Patients Performed on St-Genesia Show Hypercoagulable State

**DOI:** 10.3390/jcm11247255

**Published:** 2022-12-07

**Authors:** Beverly Buffart, Anne Demulder, Marco Fangazio, Laurence Rozen

**Affiliations:** Department of Hematology, LHUB-ULB, Université Libre de Bruxelles (ULB), 1020 Brussels, Belgium

**Keywords:** COVID-19, hypercoagulable state, thrombin generation, thromboembolic risk, thrombomodulin

## Abstract

Background: At the dawn of the pandemic, severe forms of COVID-19 were often complicated by thromboembolisms. However, routine laboratory tests cannot be used to predict thromboembolic events. The objective of this study was to investigate the potential value of the thrombin generation test (TGT) in predicting hypercoagulability and thrombotic risk in the aforementioned set of patients. Methods: The study panel comprised 52 patients divided into two groups (26 COVID-19 positive and 26 COVID-19 negative); COVID-19-positive patients were further grouped in “severe” (*n* = 11) and “non-severe” (*n* = 15) categories based on clinical criteria. The routine blood tests and TGT of these patients were retrospectively analyzed. Results: All 26 COVID-19-positive patients showed decreased lymphocyte, monocyte and basophil counts and increased lactate dehydrogenase (LDH), aspartate aminotransferase (AST), and alanine transaminase (ALT) compared with control patients. Conversely, we did not observe statistically significant differences between severe and non-severe patients despite anecdotal variations in the distribution patterns. TGT without thrombomodulin (TM) addition showed statistically significant differences in the thrombin peak heights between COVID-19-positive and negative patients. After addition of TM, peak height, Endogenous Thrombin Potential (ETP) and velocity index were increased in all COVID-19-positive patients while the percentage of inhibition of ETP was reduced. These trends correlated with the severity of disease, showing a greater increase in peak height, ETP, velocity index and a drastic reduction in the percentage of ETP inhibition in more severely affected patients. Conclusions: Our data suggest that all COVID-19 patients harbor a hypercoagulable TGT profile and that this is further pronounced in severely affected patients.

## 1. Introduction

During the initial phases of the COVID-19 pandemic, patients showed great variability in clinical presentations, which ranged from asymptomatic to severe disease with high mortality [1,2]. The latter was often complicated by thrombotic events. Laboratory findings showed increased D-dimer and fibrinogen degradation products, increased fibrinogen levels, moderately decreased platelet levels and, as the disease progressed, moderately prolonged prothrombin time (PT). Most patients with high risk factors for the development of thrombotic complications were placed under antithrombotic prophylaxis with unfractionated heparin or low-molecular-weight heparins [3,4,5]. Given the well-documented inadequacy of routine coagulation tests to predict thrombotic risk, we wanted to determine whether the TGT could provide added value for the thrombotic risk assessment of COVID-19 patients.

## 2. Materials and Methods

Patients (*n* = 52) were enrolled in the study upon arrival at the emergency department of the Brugmann university hospital, Saint-Pierre university hospital and Erasme university hospital between 16 April 2020 and 11 March 2021. Blood samples were collected in 3.2% sodium-citrate tubes in order to perform routine coagulation and TGT tests, in potassium-EDTA tubes for blood count tests and in lithium-heparin tubes for biochemical analysis. Patients under anticoagulant treatment were excluded from the study. Routine coagulation tests such as PT, international normalized ratio (INR), activated partial thromboplastin time (aPTT), fibrinogen and D-dimers were assessed on fresh platelet-poor plasma obtained by standard methods of centrifugation using a Sysmex^®^ CS-5100 analyzer (Siemens, Germany), on which QC (Lyphocheck coagulation control, Bio-rad, United States) was run 2× per day. Plasma samples intended for TGT analysis were frozen at −80 °C after double centrifugation at 1900× *g* for 15 min, and upon thawing in a water bath at 37 °C for 15 min, they were analyzed on a St-Genesia analyzer (Diagnostica Stago, Asnières, France). TGT was performed with and without TM addition using thromboscreen reagent that contains an intermediate level of tissue factor. Reference plasma (Diagnostica Stago, Asnières, France) was used to normalize the results. Whole blood counts were performed on Sysmex^®^-XN-9000 analyzer (Sysmex Europe GMBH, Germany), on which QC (XN Check, Sysmex Europe, GMBH Germany) was run 2× per day. Routine biochemistry (AST, ALT, LDH and c-reactive protein (CRP)) was performed on Cobas^®^-8000 analyzer (Roche Diagnostics International^®^, Rotkreuz, Switzerland), on which QC (Liquid Assayed Multiqual, Bio-rad, United States) was run 2× per day.

Patients were retrospectively classified as COVID-19 positive or COVID-19 negative after review of their medical records. COVID-19-positive patients had a positive PCR test and a highly suggestive lung CT scan. The group of COVID-19-positive patients was further divided into two subgroups (severe and non-severe) based on clinical presentation and clinical course; notably, the non-severe group included patients who did not required hospitalization, whereas the severe group patients presented hypoxemic pneumopathy requiring oxygen therapy and/or damage involving >25% of the lung parenchyma according to the CT scan. The COVID-19-negative group had heterogeneous symptomatology, including major inflammatory diseases but tested negative at the SARS-CoV-2-specific PCR and/or did not have CT scan images suggestive of COVID-19 pneumopathy. Patients belonging to the two groups, COVID-19 positive and negative, were matched for gender and age.

The TGT data analysis included: lag time, peak height, time to peak, ETP, ETP inhibition, velocity index and start tail. ETP inhibition was calculated based on comparison of the test results with and without TM. Their ratio was used to estimate the percent of anticoagulation due at the protein C/protein S (PC/PS) system.

Categorical variables were compared by the χ^2^ test when appropriate (Microsoft^®^ Excel, Microsoft, Redmond, WA, USA). Continuous variables were compared by the Mann–Whitney test (GraphPadPrism^®^ software, GraphPad Software, San Diego, CA, USA—version 7.00, 2016). All statistical tests were two-sided. Statistical significance was defined as *p* ≤ 0.05.

## 3. Results

### 3.1. Characteristics of the Study Population

52 patients were enrolled in the study and divided into two groups: 26 COVID-19-positive patients and 26 COVID-19-negative patients. Each group included 14 females and 12 males; the median age was 52 years (47–71 years) for the COVID-19-positive group and 51 years (47–71 years) for the COVID-19-negative group. Among the COVID-19-positive group, 11 patients presented a severe disease, and 15 patients had a non-severe disease (6 and 8 females and 5 and 7 males, respectively). The median ages were 50 years (48–74 years) and 53 years (46–71 years), respectively.

### 3.2. Routine Results

Statistically significant differences were observed between COVID-19-positive and COVID-19-negative patients for white blood cells count, lymphocyte count, monocyte count, eosinophil count, basophil count, LDH, AST and ALT (Table 1). Lymphocyte count was below the reference value for the COVID-19-positive group, while white blood cell count, monocyte count, eosinophil count, and basophil count were within the reference values but showed a decreasing trend. LDH was elevated in all the COVID-19 patients. Whilst remaining within the reference values, AST and ALT tended to be higher in the COVID-19-positive group. No statistically significant differences between COVID-19-positive and COVID-19-negative patients were observed for PT, INR, aPTT, fibrinogen, D-dimers, platelets, neutrophils or CRP (Table 1). 

Within the COVID-19-positive group, the median value of CRP, D-dimers and LDH exceeded the reference values, while the lymphocyte count was below the reference values regardless of the severity of the clinical condition of the patients. The AST median value was above the reference values only for the severe subgroup. Conversely, the fibrinogen median value was elevated only in the non-severe subgroup. Despite these trends, there was no statistically significant difference based on the disease severity.

### 3.3. Thrombin Generation Results

TGT results without TM addition showed an increased normalized peak height in COVID-19-positive patients compared with COVID-19-negative patients (Table 2, Figure 1). This observation correlated with the severity of the disease as shown by the statistically significant difference between the severe and non-severe subgroups. The highest values of normalized velocity index were found in the COVID-19-positive group and particularly in patients affected by a more aggressive disease, although it did not reach a statistically significant value.

Upon TM addition, the peak height, ETP and velocity index were significantly higher while ETP inhibition was reduced in COVID-19-positive patients compared with patients not affected by the disease (Figure 1). These differences correlated with the severity of the disease, as more extreme values of peak height, endogenous thrombin potential, velocity index and ETP inhibition were observed in severely affected patients (Figure 2). We performed ROC curves (Appendix A) without add-value to discriminate between COVID-19-positive and COVID-19-negative patients because we could not establish a cut-off value that would combine sufficient specificity and sensitivity. However, better discrimination was observed between COVID-19 “severe” and COVID-19 “non-severe”/COVID-19 negative. A larger cohort would probably have allowed better discrimination between the different groups.

## 4. Discussion

Gautret et al., 2020 [6] and Luo et al., 2020 [7] showed that men with co-morbidities (hypertension, diabetes or coronary heart disease) and over the age of 65 were more severely affected by the disease. Conversely, our cohort of patients did not present statistically significant differences in gender and age between the severe and non-severe COVID-19-positive subgroups. Nevertheless, during the study, three patients died from complications of COVID-19. These patients had multiple comorbidities (hypertension, diabetes, BPCO, overweight and a history of deep vein thrombosis). In addition, two severe COVID-19-positive patients developed thrombotic complications, more specifically, pulmonary embolisms. They were subsequently treated with LMWH. In addition to these two patients, one severe COVID-19 patient also received LMWH during hospitalization to prevent thrombotic events.

All 26 COVID-19-positive patients showed lymphopenia and lower platelet count in line with the literature [6,8,9,10]. Coagulation tests (PT and APTT) were normal in our cohort, which is in disagreement with previous studies that described PT abnormalities in patients affected by a severe form of the disease [11]. Many studies have shown that high D-dimers levels are associated with severity of the disease and mortality [12,13]. In our cohort, the highest fibrinogen values were observed in COVID-19-positive patients, although it did not reach statistical significance compared with controls. Less severe patients had higher fibrinogen levels than severely affected patients. This might correlate with the hypercoagulable state found in severely affected patients, and it might be an indicator of the unbalance of the coagulation cascade resulting, in a subset of these patients, in fibrinogen consumption and thus a decrease in its plasmatic levels. Thachil et al. [14] suggested that fibrinogen may have a protective role against viral infection by helping to regulate inflammation. The literature reports conflicting results; some studies showed high fibrinogen and D-dimers in the most severely affected patients [15] whereas others found high D-dimers values associated with low fibrinogen levels in non-surviving patients [16]. This observation was associated with disseminated intravascular coagulation in the severe forms of the disease [17]. 

In agreement with the literature, COVID-19-positive patients presented higher CRP levels compared with the COVID-19-negative group, with the highest levels reached in severely affected patients [7]. LDH levels were elevated in all COVID-19-positive patients with no clear difference between severe and non-severe patients. The liver enzymes AST and ALT were elevated in the COVID-19-positive group, with a marked difference observed for AST. High LDH [18,19], AST and ALT [20,21] levels were associated with higher risk of poor outcome [22].

TGT results displayed a hypercoagulable profile for SARS-CoV-2-infected patients compared with control patients. In severely affected patients, the profile more strikingly presented the characteristics of a hypercoagulable state. This was also characteristic of the two patients who developed pulmonary embolisms during their hospitalization. To date, only a few studies addressed the issue of thrombin generation in patients with COVID-19. White et al. [15] compared critical to non-critical COVID-19 patients and found lower peak heights with and without TM addition in severely affected patients. This finding must be interpreted in light of the fact that patients under prophylactic or therapeutic anticoagulation were included in the study. The authors treated plasmas with heparinase or DOAC-Remove^®^ (5-Diagnostics) in order to limit anticoagulant interference, but this approach seems questionable to us. Campello et al. [23] also found that peak heights were lower in patients admitted to intensive care units for COVID-19 compared with patients with mild COVID-19, but they did not disclose their thromboprophylaxis status. They subsequently performed a sub-analysis comparing healthy patients with COVID-19-positive patients treated or not with anticoagulants. In agreement with our results, peak heights were higher in untreated COVID-19 patients. 

Our data suggest a strong decrease in inhibition of ETP after addition of TM in COVID-19-positive patients and an even greater effect in the subgroup of severely affected patients. This observation is in line with the aforementioned results of Campello (COVID-19 positive patients with or without thromboprophylaxis) and with the analysis of de la Morena-Barrio, who compared healthy individuals with patients affected by pneumonia and observed that the addition of TM did not decrease ETP in patients independently of the etiology (SARS-CoV-2 or others). Although a significant proportion of patients were anticoagulated, these data suggest a dysfunction of the anticoagulant PC/PS system [23,24] (Table 3).

In order to avoid this bias, in our study, patients under anticoagulant therapy were excluded. Consequently, we reinforce the suggestion that the PC/PS system fails in patients affected by COVID-19 proportionally to the severity of the disease and, presumably, to the release of pro-inflammatory cytokines.

Given our strict enrollment criteria of non-anticoagulated patients prior blood collection, the number of samples included in our study is limited. Furthermore, the retrospective nature of the study forced us to perform the TGT only in the absence of TM for the samples for which the collected plasma was insufficient. This prevented us from analyzing these plasmas with or without the addition of the PC/PS cofactor, which ultimately rendered impossible the important comparison of the TGT with or without TM. For all these reasons, the data analysis could benefit of further studies for validation with a larger and independent cohort of patients.

## 5. Conclusions

We showed that the use of a global hemostasis test as the TGT on the ST-Genesia analyzer detects a hypercoagulability state in COVID-19-positive patients. This hypercoagulability is linked to a dysfunctional PC/PS system, as proven by the results observed upon addition of TM to the tested plasmas and correlates with the severity of the disease. Prospective studies aimed at defining a TGT COVID-19 patient stratification according to the coagulability state could help determine if different prophylactic and/or treatment measures might improve the outcome of the patients. 

Whether hypercoagulability is still present in long COVID patients and in patients affected by a disease caused by one of the newer viral variants remains an open question.

## Figures and Tables

**Figure 1 jcm-11-07255-f001:**
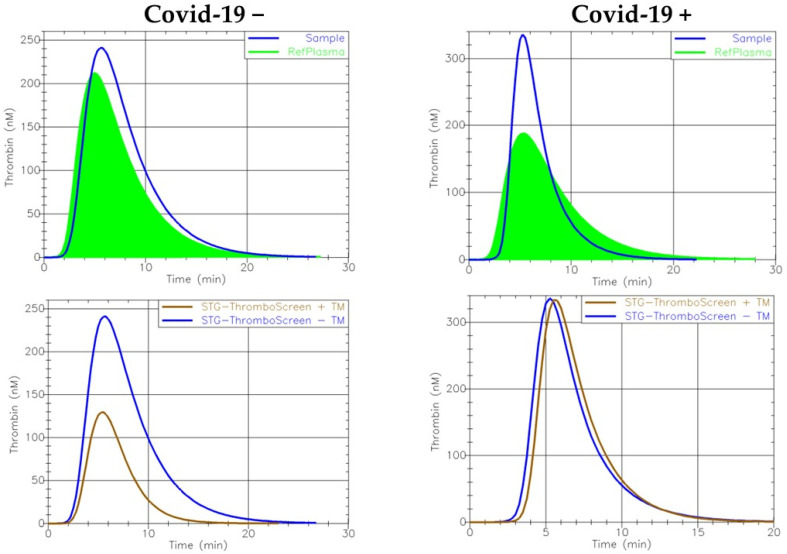
Example of thrombograms with and without TM obtained by ST-Genesia analyzer in a COVID-19-positive patient and a COVID-19-negative patient.

**Figure 2 jcm-11-07255-f002:**
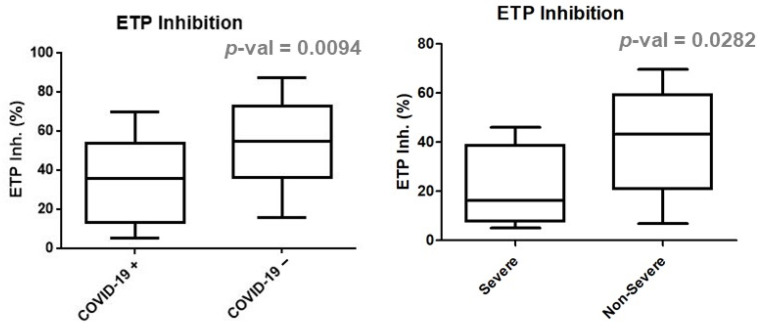
Comparison of inhibition of ETP (%) in COVID-19-negative and COVID-19-positive patients as well as COVID-19-positive “severe” and “non-severe” subgroups.

**Table 1 jcm-11-07255-t001:** Laboratory results of COVID-19-negative and COVID-19-positive patients as well as COVID-19-positive “severe” and “non-severe” subgroups.

	Reference Values	*n* = COVID-19 + *n* = COVID-19 −	COVID-19 +	COVID-19 −	*p*-Value	*n* = Severe *n* = Non-Severe	Severe	Non-Severe	*p*-Value
Platelets (×10^3^/µL)	150–440	*n* = 26 *n* = 25	221 (186–276)	235 (201–334)	0.2139	*n* = 11 *n* = 15	202 (180–254)	221 (189–281)	0.5149
White blood cells (×10^3^/µL)	3.50–11.00	*n* = 26 *n* = 25	6.43 (5.05–8.81)	8.57 (6.69–11.21)	0.0276	*n* = 11 *n* = 15	6.27 (6.1–6.97)	6.44 (4.23–8.86)	>0.9999
Neutrophils (×10^3^/µL)	1.50–6.70	*n* = 23 *n* = 22	4.63 (3.06–7.31)	6.55 (4.48–8.89)	0.1335	*n* = 10 *n* = 13	4.91 (4.03–6.07)	3.95 (2.89–7.99)	0.5334
Lymphocytes (×10^3^/µL)	1.20–3.50	*n* = 23 *n* = 22	1.01 (0.61–1.36)	1.43 (1.15–1.67)	0.0074	*n* = 10 *n* = 13	1.06 (0.57–1.80)	1.01 (0.76–1.24)	0.9757
Monocytes (×10^3^/µL)	0.20–1.00	*n* = 23 *n* = 22	0.36 (0.28–0.59)	0.58 (0.43–0.98)	0.0102	*n* = 10 *n* = 13	0.39 (0.32–0.53)	0.36 (0.27–0.62)	>0.9999
Eosinophils (×10^3^/µL)	<0.40	*n* = 23 *n* = 22	0.01 (0.00–0.05)	0.07 (0.02–0.26)	0.0058	*n* = 10 *n* = 13	0.01 (0.00–0.06)	0.02 (0.00–0.08)	0.9140
Basophils (×10^3^/µL)	< 0.10	*n* = 23 *n* = 22	0.01 (0.01–0.02)	0.03 (0.02–0.04)	0.0033	*n* = 10 *n* = 13	0.01 (0.01–0.02)	0.01 (0.01–0.03)	0.8621
CRP (mg/L)	<5.0	*n* = 25 *n* = 25	12.4 (4.4–84.3)	7.3 (1.6–29.0)	0.0815	*n* = 11 *n* = 14	36.0 (4.8–150.0)	6.9 (4.4–50.4)	0.2493
PT (%)	70–130	*n* = 26 *n* = 26	103.9 (86.0–109.3)	94.6 (85.0–100.8)	0.1065	*n* = 11 *n* = 15	107.3 (86.3–115.7)	97.3 (85.0–109.0)	0.5143
INR	0.95–1.31	*n* = 26 *n* = 26	0.99 (0.97–1.07)	1.03 (1.00–1.07)	0.1421	*n* = 11 *n* = 15	0.98 (0.94–1.08)	1.02 (0.97–1.07)	0.6356
APTT (s)	21.6–28.7	*n* = 26 *n* = 26	23.9 (22.2–25.7)	24.3 (22.6–26.1)	0.5642	*n* = 11 *n* = 15	23.8 (22.2–25.8)	24.4 (22.1–25.6)	0.8484
Fibrinogen (mg/dL)	150–400	*n* = 14 *n* = 8	415 (330–522)	333 (295–633)	0.5585	*n* = 5 *n* = 9	391 (323–456)	466 (320–526)	0.4171
DDIM (ng/mL)	<500	*n* = 23 *n* = 13	727 (370–2827)	486 (310–1094)	0.2427	*n* = 11 *n* = 12	875 (447–3084)	572 (359–2553)	0.5769
LDH (UI/L)	♂135–225 ♀135–214	*n* = 22 *n* = 25	316 (237–396)	193 (169–221)	<0.0001	*n* = 10 *n* = 12	351 (245–536)	294 (221–339)	0.0965
AST (UI/L)	♂ < 40 ♀ < 32	*n* = 25 *n* = 25	30 (24–53)	20 (14–26)	0.0005	*n* = 11 *n* = 14	40 (27–58)	28 (21–53)	0.1877
ALT (UI/L)	♂ < 41 ♀ < 3 3	*n* = 25 *n* = 26	23 (18–41)	17 (12–23)	0.0191	*n* = 11 *n* = 14	28 (18–56)	20 (16–36)	0.3366

**Table 2 jcm-11-07255-t002:** TGT results with and without TM for COVID-19-negative and COVID-19-positive patients as well as COVID-19 positive “severe” and “non-severe” subgroups.

	**Normal Values**	**COVID-19 +** **(*n* = 26)**	**COVID-19 −** **(*n* = 26)**	***p*-Value**	**Severe** **(*n* = 11)**	**Non-Severe** **(*n* = 15)**	***p*-Value**
Without TM							
Normalized lag time	0.9–2.0	1.30 (1.06–1.55)	1.23 (1.08–1.53)	0.7680	1.33 (1.10–1.39)	1.27 (0.99–1.61)	>0.9999
Normalized peak height (%)	26.4–185.6	130.1 (120.2–150.4)	111.5 (83.7–132.9)	0.0171	146.6 (129.0–166.7)	123.8 (100.4–136.6)	0.0316
Normalized time to peak	0.8–1.8	1.12 (0.96–1.28)	1.22 (0.99–1.47)	0.3235	1.10 (0.99–1.21)	1.14 (0.87–1.46)	0.7114
Normalized ETP (%)	47.3–187.2	109.4 (96.9–132.7)	106.9 (93.4 –119.1)	0.4399	114.3 (103.3–137.3)	105.3 (83.3–122.1)	0.2586
Normalized velocity index (%)	14.4–212.4	133.2 (103.6–182.3)	105.0 (67.4–138.1)	0.0584	167.6 (117.2–226.2)	121.9 (87.9–168.9)	0.1639
Normalized start tail (min)	/	0.87 (0.78–1.00)	0.96 (0.83–1.22)	0.0932	0.87 (0.81–0.97)	0.90 (0.73–1.03)	0.8277
	**Normal Values**	**COVID-19 +** **(*n* = 23)**	**COVID-19 −** **(*n* = 24)**	***p*-Value**	**Severe** **(*n* = 8)**	**Non-Severe** **(*n* = 15)**	***p*-Value**
With TM							
Lag time (min)	1.9–4.7	2.77 (2.28–3.36)	2.84 (2.49–3.45)	0.5582	2.72 (2.31–3.24)	2.77 (2.28–3.51)	0.7402
Peak height (nM)	30.3–409.5	218.8 (160.5–280.8)	134.1 (80.2–215.4)	0.0090	307.5 (229.9–341.4)	173.0 (138.5–225.3)	0.0042
Time to peak (min)	3.7–7.2	4.71 (3.98–5.65)	4.99 (4.29–6.10)	0.3607	4.70 (4.12–5.16)	4.81 (3.94–6.14)	0.7763
ETP (nM.min)	245.5–2078.0	912.1 (795.4–1209.0)	656.0 (398.7 –899.5)	0.0067	1274.0 (1064.0–1583.0)	830.7 (578.3 –932.4)	0.0042
Velocity index (nM/min)	8.9–283.7	143.1 (85.6–218.3)	79.2 (51.9–150.9)	0.0138	222.4 (142.7–246.0)	104.6 (77.7–180.7)	0.0194
Start tail (min)	/	14.47 (13.7–16.28)	14.90 (13.58–15.81)	0.9790	14.97 (13.86–16.01)	14.47 (13.59–16.28)	0.7648
Inhibition ETP (%)	14.4–75.2	35.68 (13.24–53.64)	54.97 (36.07–72.79)	0.0094	16.26 (7.64–38.69)	43.15 (20.76–59.24)	0.0282

**Table 3 jcm-11-07255-t003:** Comparison of various studies that treated TGT in COVID-19 patients.

Study	Design	TGT Results	Conclusion	Critical Review
De la Morena-Barrio et al. [24]	Retrospective analysis 1° COVID-19 vs. healthy subjects 2° COVID-19 vs. SARS-CoV-2-negative pneumonia 127 hospitalized COVID-19-positive patients (78% with antithrombotic therapy) 24 hospitalized patients with SARS-CoV-2-negative pneumonia (54.2% with antithrombotic therapy) 12 healthy subjects All the subjects >18 years	1° Peak height was higher in COVID-19 patients than healthy subjects. (*p* = 0.011) No statistical difference for ETP Ratio ETP (with TM/without TM) was higher in COVID-19 patients (*p* = 0.023) 2° Peak height was higher in SARS-CoV-2-negative pneumonia than COVID-19 patients (*p* = 0.037) ETP was higher in SARS-CoV-2-negative pneumonia patients (*p* = 0.005) No statistical difference for Ratio ETP (with TM/without TM) The study also correlated low ETP with poor prognosis and the occurrence of complications.	“Despite the frequent use of heparin, COVID-19 patients had similar thrombin generation to healthy controls”	These results must be interpreted with great care given the known interference of anticoagulants with TGT. The use of heparin may well explain the decrease in thrombin generation in COVID-19 patients who then have a similar profile to healthy patients. For the same reason, it could be that low ETP associated with complications are due to the use of heparin.
White et al. [15]	Retrospective analysis 34 patients with noncritical COVID-19 (94% with anticoagulation) 75 patients with critical COVID-19 (94% with anticoagulation) All the subjects >18 years	No statistical difference for peak height, ETP or ETP Inhibition	“Disease severity did not increase thrombin generation when comparing both cohorts; counter-intuitively critical patients were hypocoagulable”	The results should be interpreted with caution given the anticoagulation of almost all subjects. To avoid interference, they treated the samples with DOAC-remove, but this approach remains questionable. The wide use of anticoagulant in the two groups compared in the study may explain why they do not observe a statistical difference for the parameters mentioned. Thrombin generation will tend to decrease in the presence of anticoagulant
Campello et al. [23]	Prospective analysis 1° COVID-19 vs. healthy subjects 2° Mild disease vs. ICU 3° Healthy vs. COVID-19 without thromboprophylaxis 4° COVID-19 with thromboprophylaxis vs. COVID-19 without thromboprophylaxis 89 COVID-19 patients (59 «mild disease» (59.3% with thromboprophylaxis) and 30 “ICU patients”(100% with thromboprophylaxis)) 54 healthy subjects All the subjects >18 years	1° No statistical difference for peak height, ETP or ETP Inhibition 2° Peak height was higher in mild disease (*p* = 0.010) ETP was higher in mild disease (*p* = 0.012) ETP Inhibition was higher in ICU COVID-19 (*p* = 0.003) 3° Peak height was higher in COVID-19 patients without thromboprophylaxis than in healthy subjects (*p* < 0.01) ETP was higher in COVID-19 patients without thromboprophylaxis than in healthy subjects (*p* < 0.01 without TM, *p* < 0.0001 with TM) ETP inhibition was significantly decreased in COVID-19 patients without thromboprophylaxis than in healthy subjects. (*p* < 0.001) 4° Peak height was higher in COVID-19 patients without thromboprophylaxis than in COVID-19 patients with thromboprophylaxis (*p* < 0.001) ETP was higher in COVID-19 patients without thromboprophylaxis than in COVID-19 patients with thromboprophylaxis (*p* = 0.009 without TM, *p* = 0.0003 with TM) ETP Inhibition was significantly decreased in COVID-19 patients without thromboprophylaxis (*p* = 0.0003)	“In conclusion, our study showed that patients with COVID-19 had increased TG at diagnosis and confirmed that standard dose thromboprophylaxis could at most reduce TG to the levels of healthy controls. Intermediate sub-therapeutic LMWH dose more effectively inhibited TG in patients with severe COVID-19 by increasing ETP inhibition via ETP with TM reduction.”	If we focus on the parameters of the TGT that took into account whether or not thromboprophylaxis was taken, we observe that the obtained results are in favor of hypercoagulability in COVID-19 patients. The decreased ETP inhibition in COVID-19 patients without thromboprophylaxis supports our idea of a failure of the PS/PC system in COVID-19 patients.

## Data Availability

The data presented in this study are available on request from the corresponding author. The data are not publicly available due to patients’ privacy.

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
