# Peer review of "Global Hemostasis Potential in COVID-19 Positive Patients Performed on St-Genesia Show Hypercoagulable State"

_jcm, 2022, doi:10.3390/jcm11247255_

Round 1
Reviewer 1 Report
Thank you for the invitation to review this paper. This is an interesting paper, it is aimed to evaluating if TGT could add some value in the diagnosis of thrombotic risk associated with COVID-19 infections. The study presented used a population of 26 affected patients (11 severe and 12 non-severe) and 26 controls that come with some afflictions but not positive COVID test. The controls were age and gender matched, but it is uncertain if these controls are appropriate controls since it is not clear what their status is, are they considered healthy or something else since they have some afflictions. Regardless of that, this study uses a very small sample which brings some concern on statistical power. The paper does not specify or describe what would be “acceptable value” therefore it is difficult to understand its significance.
The paper is missing many crucial and very important information points. There are no dates of data collection, there is no power analysis, IRB review (although it says consents were obtained), there is no way to know which COVID strain is being evaluated. All these factors are unaccounted and may add important caveats to the study.
The statistical approach is a valid way to compare groups but is not optimal for the objective presented, why not using ROC analysis? For evaluating diagnostic potential, knowing sensitivity and specificity are the norm. A scheme that could be used will require 2 comparison groups:
· (ROC analysis of COVID-) vs. (COVID+ non severe) + (COVID+ severe)
· (ROC analysis of COVID-) + (COVID+ non severe) vs. (COVID+ severe)
In the discussion, there appears to be some confusing arguments, there is a mention that White et al. found lower peak height in severe cases but research here say the opposite. Table 2 and Figure 1 show that peak height is higher in the COVID+ group. Please tell me if I am getting this wrong.
At the end, I am skeptical on what is the significance and relevance of this paper. This paper appears to be done using the original COVID 19 strain, that strain is not represented in the population anymore. In their closing argument in their conclusion, they do mention long COVID and new variants being something to pursue in the future. So why publish this on the original strain now? The original strain is not represented in the population anymore, besides being a reference point for the future studies there is little value for this work.
Reviewer 2 Report
The research topic is relevant. and is necessary to understand the formation of thrombosis in patients with confirmed Covid 19.
The proposed methods and a set of tests are universal and widely used in the treatment of patients. The data obtained during the study complement and expand the study of viral diseases.
1. In the methodological part, it is necessary to indicate the use of third-party quality controls for obtaining results (Rendox, Bio-Rad or analogues) proving the reliability of the results obtained.
2. Detail the research issues and update the results obtained. In the discussion, expand the significance of the results obtained for the clinical work of doctors and take into account the results obtained in practice in the treatment of patients with Covid.
3. The material does not provide data on the conclusion of the ethical committee of the hospital on the use of data for a scientific article.
Reviewer 3 Report
The authors try to bring more light in the understanding of Covid-19 associated hypercoagulability.
There are several weaknesses in the paper; major issues to be considered:
In the abstract (e.g. "however current laboratory tests are not comprehensive enough to predict thromboembolic event"s) and in the introduction the authors raise the expectation to identify patients at highest risk of VTE, but this is not shown in the paper, even the VTE-complications in these patients are not given.
This paper may help to unterstand and demonstrate hypercoagulability in Covid-19+ patients, but the value of hemostasis testing in Covid-19 + patients with St-Genesia remains open..
I do not nderstand why patientiens are obviously classified "Covid-19+" without a confirmative (obtained later on) PCR.
The numbersof patients given in the paper are very well balanced between the groups, not very convincing that consecutive patients in the emergency department were entered, but rather suggestive that there was some kind of choosing fitting patients.
Furthermore, hospitalization (the discriminating factor between severe vs non-severe) may depend more on comorbidities (with different degrees of hypercoagulability independent of Covid-19) than on severity of Covid-19. Therefore more background information of the patients in the study should be provided; including post administration anticoagulation, bleedings and thromboembolic events.
Within the discussion terms such as "higher" need to be based on significance in the statistic evaluation (reports).
Reviewer 4 Report
I would recommend to use the words "positive" and negative in the title and abstract instead of the "+" and "-" for a better understanding.
Also, i would like to see abbreviations such as: Covid-19 - meaning Covid-19 negative etc.
The word "Covid" should be followed by "-19" to avoid errors [line 30].
Punctuation signs are missing [line 32].
In the Discussions area I would like to see more comparisons between the results that the authors present and other articles from literature that follow the same methodology.
Also, a table with the comparisons would be highly recommended.
The Conclusion part should be more pertinent and impersonal.
Overall, the whole paper needs an upgrade in the quantitative and qualitative English language.
References must be modified accordingly to the example in the authors' guide.
Round 2
Reviewer 1 Report
I mush admit that I am very surprised with how much improvement was done in the paper. Great job by the authors.
The addition of table 3 is very significant and makes a big difference in the clarity of the study.
The methods are also greatly improved. Although they could still be improved, they are now at a sufficient level of description for publication.
The authors should consider adding the ROC they provided in the response to the main paper or at least making it available as supplementary. In addition to the ROC analysis of COVID-) + (COVID+ non severe) vs. (COVID+ severe), please add as well the (ROC analysis of COVID-) vs. (COVID+ non severe) + (COVID+ severe), these two ROCs can objectively present the potential application the main argument.
I would support the publication of this paper provided that the ROC curves are added.
Reviewer 3 Report
The revised paper is clearly improved.
The reviewer would be "happy", if the authors would cleary state, that their test system - as demonstrated - detects "hypercoagulability" in Covid-19-pts., but that - as long as a prospective intervention study in missing - it is absolutely open, if usage of this test in routine patient care would help to improve patient outcome.
Reviewer 4 Report
Congratulations!
The changes are in line with the requests.
